# Helicopter Planet Gear Rim Crack Diagnosis and Trending Using Cepstrum Editing Enhanced with Deconvolution

**DOI:** 10.3390/s24082593

**Published:** 2024-04-18

**Authors:** Nader Sawalhi, Wenyi Wang, David Blunt

**Affiliations:** Defence Science and Technology Group (DSTG), Melbourne, VIC 3207, Australia; nader.sawalhi@defence.gov.au (N.S.); david.blunt@defence.gov.au (D.B.)

**Keywords:** planet gear, rim crack, cepstrum editing, liftering, minimum entropy deconvolution (MED), signal synchronous average (SSA)

## Abstract

Detecting gear rim fatigue cracks using vibration signal analysis is often a challenging task, which typically requires a series of signal processing steps to detect and enhance fault features. This task becomes even harder in helicopter planetary gearboxes due to the complex interactions between different gear sets and the presence of vibration from sources other than the planetary gear set. In this paper, we propose an effectual processing algorithm to isolate and enhance rim crack features and to trend crack growth in planet gears. The algorithm is based on using cepstrum editing (or liftering) of the hunting-tooth synchronous averaged signals (angular domain) to extract harmonics and sidebands of the planet gears and low-pass filtering and minimum entropy deconvolution (MED) to enhance extracted fault features. The algorithm has been successfully applied to a vibration dataset collected from a planet gear rim crack propagation test undertaken in the Helicopter Transmission Test Facility (HTTF) at DSTG Melbourne. In this test, a seeded notch generated by an electric discharge machine (EDM) was used to initiate a fatigue crack that propagated through the gear rim body over 94 load cycles. The proposed algorithm demonstrated a successful isolation of incipient fault features and provided a reliable trending capability to monitor crack progression. Results of a comparative analysis showed that the proposed algorithm outperformed the traditional signal processing approach.

## 1. Introduction

Gears are critical transmission components that are utilized widely in industry due to their reliability and high torque transmission capacity. However, gears are often subjected to high torques and cyclic loads, which eventually lead to component failures due to fatigue. Gearbox health monitoring is conducted using a variety of methods including wear debris monitoring [1,2], thermography [3] and vibration analysis [4,5,6]. It is common to combine more than one method, e.g., wear debris and vibration trending, to provide a better indication of health through a hybrid model [7]. Planetary gearboxes, being compact and able to handle high torques, have been used widely in transmission systems like helicopter main gearboxes and wind turbines. For a planetary gearbox, accelerometers are typically placed radially on the casing outside the ring gear to capture vibration data. Vibration signals collected through these sensors contain a mixture of harmonics and sidebands in their spectra due to the multiple sources of vibration from each gear mesh (planet-sun and planet-ring), each with the same meshing frequency. This further becomes complicated by varying transfer paths to the location of the fixed accelerometer and the potential differences in phase between the individual gear meshes [8,9]. This complex interaction requires the use of advanced signal processing/pre-processing tools, which attempt to isolate/separate individual time-varying components [10,11]. The use of some pre-processing techniques such as synchronous signal averaging (SSA) [12], discrete random separation (DRS) [13] and linear prediction filtering (LPF) [14], often do not result in clear fault diagnosis, even when there are only small speed fluctuations and the system can be considered to be a mathematically stationary process.

Challenging (to detect) failure modes in planet gears can happen due to fatigue crack initiation and progression in either the tooth or the gear rim. Typically, fatigue cracks do not generate much wear debris and, as such, vibration analysis is often relied upon to diagnose the presence of the crack at an early stage and trend its progression. Rim fatigue cracking in planet gears was identified as the main cause of the major crashes of an AS-332L2 in 2009 [15] and an EC 225 LP in 2016 [16], both from the Super Puma family of helicopters. A tooth crack mostly happens at either the root or the tip of the gear tooth as a result of bending fatigue or impact load. Rim cracking takes place at the rim or web of the gear as a result of excessive stress concentration. A rim crack will cause a periodic reduction of mesh stiffness of a gear pair, which shows as impulses/impacts in the vibration signal when the crack passes through the load path. This results in an increase in sidebands around the gear mesh frequency and its harmonics. These impulses/impacts will not be clear in the raw vibration signal until a very late stage of crack development, but they can be detected earlier in the residual signals (i.e., after removing strong regular gear mesh components). Although extensive research efforts and work have been devoted to the early detection of gear cracks [17,18,19,20,21,22], more work is still needed to address this critical area in planet gears and provide reliable and efficient algorithms for early detection and monitoring.

In this paper, we propose a new diagnosis algorithm using a cepstrum-editing technique [23,24] applied to synchronously averaged signals over the (planet-ring) hunting-tooth period (angular domain) to extract periodic features relating to crack symptoms and then, using minimum entropy deconvolution (MED) [25] to further enhance the extracted features from the cepstrum-edited signals. This paper is organized as follows: Section 2 (Material and Methods) presents a summary of the experimental test rig (full-scale Helicopter Transmission Test Facility), crack initiation and data collection, traditional diagnostic approach for gear crack detection and a discussion of the proposed new signal processing algorithm. Section 3 (Results) discusses the obtained results, illustrating the advantages of the proposed technique in the early detection of cracks and the trending and clustering improvements it provides compared to the traditional approach. Section 4 (Discussion) sheds more light on the observations seen in the results obtained using the proposed new algorithm. In particular, this section presents a squared envelope analysis and illustrates the periodicity observed in the cepstrum domain analysis. Finally, Section 5 (Conclusions) provides a summary of the paper and the results.

## 2. Materials and Methods

### 2.1. Experimental Setup and Data Collection Scheme

The Defence Science and Technology Group (DSTG) completed an accelerated planet rim fatigue crack propagation test under controlled conditions in 2020–2022 using their Helicopter Transmission Test Facility (HTTF) with a Bell-206B-1 (OH-58) helicopter main transmission gearbox (4-planet version) (Bell Helicopters, Fort Worth, TX, USA) [26]. The seeded-fault test produced a vibration dataset repository suitable for exploring fatigue cracking in thin-rim helicopter planet gears. The data at one load setting (125% torque) were made available to the public through a Data Challenge organized in the HUMS 2023 conference (https://humsconference.com.au/index2.html, accessed on 1 February 2024). The dataset can be publicly accessed using the following URL: https://www.dst.defence.gov.au/our-technologies/helicopter-main-rotor-gearbox-planet-gear-fatigue-crack-propagation-test (accessed on 1 February 2024). Section 2.1.1, Section 2.1.2 and Section 2.1.3 present relevant information of the HTTF, the seeded crack initiation, progression and the dataset details, respectively.

#### 2.1.1. Helicopter Transmission Test Facility (HTTF)

The HTTF at DSTG Melbourne utilizes a retired Bell 206B-1 Kiowa (OH-58) helicopter main rotor gearbox, which has either 3-planet or 4-planet configurations. The 3-planet configuration has 3 equally spaced planets with all the meshes in phase. As such, the orders that are not integer multiples of 3 are suppressed. In the 4-planet version, two pairs of planet gears are placed diametrically opposite from each other, but there is a non-uniform angular distribution around the sun gear (i.e., approximately 88.6°, 91.4°, 88.6°, 91.4°). Opposite planets mesh 180° out of phase, leading to zero-sum odd orders for opposite pairs and in-phase even orders. A schematic presentation (3-planet) showing the main components of the gearbox as well as the number of teeth on each gear is presented in Figure 1. This gearbox has two speed reduction stages: a spiral pinion/bevel gear stage with a reduction ratio of 3.73:1 (71/19) and a planetary stage with a reduction ratio of 4.67:1 (99/27 + 1) giving a total reduction ratio of 17.44:1. In the HTTF, the gearbox runs with a nominal input speed of 6000 RPM, giving an output speed of approximately 344 RPM. Key frequencies of interest (both in Hz and planet orders) are provided in Table 1.

The HTTF is fitted with four single-axis accelerometers as shown in Figure 2. One of the sensors is placed at the input pinion flange (Input Pinion sensor), while the other three are placed on the upper housing flange around the ring gear as follows: Ring Front at the front of the gearbox; Ring Left at the left side of the gearbox; and Ring Rear at the rear of the gearbox. Note that the locations are approximate and relative to viewing forward of the helicopter. A once-per-rev inductive tachometer on the input shaft was used in the synchronous averaging process.

#### 2.1.2. Crack Initiation and Progression

The notched planet gear and a bottom view of the 4-planet carrier assembly are shown in Figure 3a and Figure 3b, respectively. Two electric-discharge-machined (EDM) notches can be seen in Figure 3a. The first notch (smaller notch of 1.4 mm) was introduced in June 2020. The gearbox was then run in approximately 30-min load cycles (the actual duration was slightly longer due to time required to transition from one setting to the next) as depicted in Table 2. After completing 146 load cycles over 16 separate days of rig operation between June and December 2020, the test failed to initiate a crack. As such, a second notch (larger notch of 5.4 mm) was introduced in the planet gear rim on the opposite side in October 2021, and testing continued for another 12 days of operation up to January 2022. This second notch successfully initiated a crack that propagated from one side of the gear to the other between the two gear teeth over 94 load cycles (i.e., load cycles 147 to 241), noting that the test was interrupted by COVID-19 lockdowns.

#### 2.1.3. Available Dataset Description

Only data from the final 58 load cycles (cycle 36 to cycle 94) after the second notch was inserted) of the test, up to the point where the crack propagated all the way through the gear rim, were made available in the HUMS 2023 Data Challenge. A total of 526 data records at the 125% torque load condition were collected across these load cycles from the four-sensor locations (Input Pinion, Ring Front, Ring Left and Ring Rear). The data were pre-processed using hunting-tooth synchronous signal averaging (H-SSA) with 405,405 re-sampled points per channel as follows:The H-SSA data length of 405,405 was designed such that the data length was an integer multiple number of revolutions for both the planet carrier and planet gear; hence, the carrier SSA (C-SSA) or planet SSA (P-SSA) can be derived from the H-SSA.The H-SSA was calculated from raw vibration signals over 12 planet-ring hunting-tooth periods, and each hunting-tooth period corresponded to 99 revolutions of planet gears (99 × 4095) or 35 revolutions of the planet carrier (35 × 11,583).The P-SSA can be produced by reshaping H-SSA into (4095 × 99) and finding the average across the 99 columns. Similarly, C-SSA can be produced by reshaping H-SSA into (11,538 × 35) and finding the average across the 35 columns.

### 2.2. Gear Crack Traditional Diagnosis Approach

Planet gear faults have been traditionally detected using a planet gear synchronous signal averaging (P-SSA) technique [10]. The steps involved in this processing are explained in the diagram of Figure 4. In the first step of this process, the H-SSA signal (signal provided in the Data Challenge dataset) is normalized to a zero mean and unit standard deviation. This has the effect of reducing the influence of any variation in torque, speed and temperature on vibration amplitudes, as well as improving the sensitivity to changes caused by the fault. The second step involves finding the planet synchronous signal average (P-SSA), which is obtained by reshaping the normalized H-SSA signal (4095 × 99) and finding the average (mean) across the 99 columns. In step 3, a residual signal is generated by removing the gear mesh harmonics and two closest sidebands in the order domain and low-pass filtering. This is performed as follows:Transferring P-SSA into the frequency domain (1-sided spectrum obtained by using Hilbert transform).Setting all GMF harmonics and the two closest sidebands around each harmonic to zero.Low-pass filtering the signal by zeroing all the spectral components above 3.5× GMF and, then, inverse transforming back to the time (or angle) domain and taking the real part of the signal. Note that the low-pass filtering represents a critical step, as rim cracks (compared to tooth cracks) are more likely to generate changes in relatively low frequency ranges. Low-pass filtering focuses attention on the lower-frequency-range features in the signal. This decision is based on a consideration of the expected changes in the dynamics of the gear system in the presence of a rim crack compared to, for example, a tooth crack fault, where the reverse approach may have been more appropriate.

**Figure 4 sensors-24-02593-f004:**
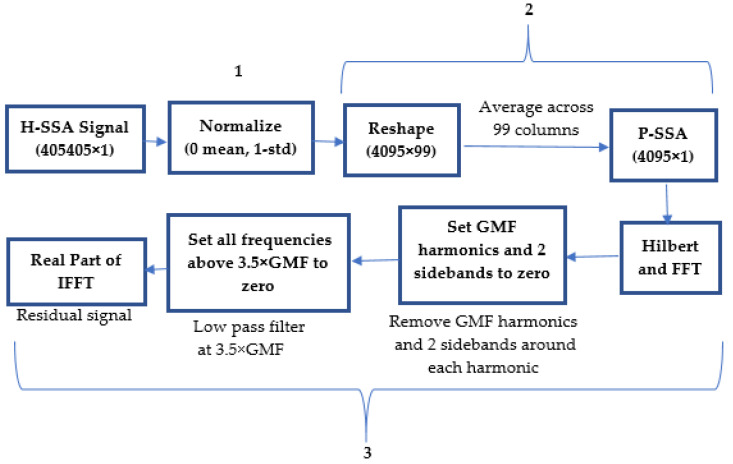
Diagram of traditional diagnosis and signal processing approach (step 1: Normalizing H-SSA; step 2: Calculating P-SSA; step 3: Deriving residual signals from P-SSA).

To illustrate the effectiveness of the traditional approach in extracting fault features, record #522 (late stage of crack growth) is examined below. Figure 5 shows the H-SSA signal of the Ring Front sensor along with the power spectral density (first four gear mesh harmonic orders that are integer multiples of 35). The signal is dominated by the gear meshing vibration and shows no obvious signs of impulses (spikes). It has a low kurtosis value of 3.26. Gear mesh harmonics can be seen clearly in the power spectrum surrounded by sidebands. A zoomed-in plot showing sidebands around the 1st and 2nd gear mesh harmonics is presented in Figure 6. It can be seen that the 1st gear mesh frequency (odd harmonic) is suppressed and that sidebands are dominant. The sidebands are spaced at 2× carrier frequency, which is half the planet pass frequency (planet pass frequency = 4× carrier frequency). A detailed explanation of GMF suppression and sidebands presence for this 4-planet gearbox was provided in [28] and compared to the 3-planet configuration. The asymmetry of the modulation sidebands about the GMF in the 4-planet arrangement was discussed based on McFadden and Smith’s model [2]. It was concluded that sideband modulation at the 2× carrier rather than at the 4× carrier happens because the planet gears in this configuration come in pairs spaced at 180° apart, and the angle between each of the planets is not 90° (approximately 88.6° and 91.4° for each pair as presented in Section 2.1.1).

Residual signals of record #522 are plotted in Figure 7. As the rim crack in the planet gear affects the meshing process with both the ring and the sun gears, two spikes (impulses) are generated during every planet revolution that are approximately 180° apart. Rim crack features (two spikes/impulses) can be seen in the signals from all four sensors, with higher amplitudes and kurtosis values seen in the Ring Left and Ring Front sensors. The higher spike in each sensor likely corresponds to the crack passing through the planet-ring mesh, and the lower to the planet-sun mesh.

### 2.3. Proposed New Signal Processing Algorithm

The proposed new signal processing algorithm to extract and enhance rim crack impulses is presented in Figure 8. The algorithm consists of four main steps, with the first and third steps being common to those used in the traditional approach (normalization and low-pass filtration). The second step works on the retrieval of planet-related harmonics and sidebands using a cepstrum-editing technique [23,29,30,31] as explained in Section 2.3.1. This step has two main purposes: the first is to retain synchronous information relevant only to planet gears, and the second is to remove the effects of the transfer path (pre-whitening). The fourth step (Section 2.3.2) employs the MED technique [25,32] to enhance the fault features (impulses).

#### 2.3.1. Cepstrum Editing

##### Cepstrum Analysis

Cepstrum analysis is a powerful nonlinear signal processing technique that was initially proposed to extract echo arrival times in composite signals. It was first introduced by Bogert et al. [33] in 1963 as the “power spectrum of a log power spectrum”. The word “cepstrum” was then devised by reversing the first syllable of ‘‘spectrum” to describe the new domain. Cepstrum was later defined by Childers et al. [34] as “the inverse Fourier transform (IFFT) of the log spectrum”. For a time domain signal x(t), the cepstrum $Ceps\left(\tau \right)$ can be written mathematically:(1)$$Ceps\left(\tau \right)=IFFT(\log X(f)),$$
where $X\left(f\right)$ is the Fourier transform (FFT) (complex) of the real time domain signal x(t), defined as follows:(2)$$X\left(f\right)=FFT\left(x\left(t\right)\right)=X_{r}(f)+{jX}_{i}(f)=A\left(f\right)exp(j\emptyset \left(f\right)),$$
where $X_{r}(f)$ and $X_{i}(f)$ are the real and imaginary components, respectively. The amplitude $A\left(f\right)$ and phase $\emptyset \left(f\right)$ are defined, respectively, as follows:$$A\left(f\right)=\left|X\left(f\right)\right|=\sqrt{{X_{r}\left(f\right)}^{2}+{X_{i}\left(f\right)}^{2}},\;\mathrm{and}$$
$$\emptyset \left(f\right)=∠X(f)=\tan^{-1}\frac{X_{i}(f)}{X_{r}(f)}.$$

Note that the cepstrum (being the inverse transfer of a log spectrum) has the units of time as its independent variable, similar to the time domain signal. Cepstra come with a set of new terminology, with quefrency, rahmonic and liftering being the three key terms to describe the cepstrum *x*-axis, cepstrum harmonics and cepstrum filtering, respectively.

If the phase in Equation (1) is retained, the logarithmic spectrum will have log amplitude as the real part and phase as the imaginary part. This so-called “complex cepstrum ${Ceps}_{c}\left(\tau \right)$” is obtained as shown in Equation (3) as follows:(3)$${Ceps}_{c}\left(\tau \right)=IFFT(\log A\left(f\right)+j\emptyset \left(f\right)).$$

The complex cepstrum has a real value (despite its name) since the log amplitude is even, while the phase is odd. Its calculation requires the phase to be a continuous function of frequency, which is often possible for frequency response functions (transfer functions). However, this is not possible for signals that consist of a mixture of deterministic discrete frequency components. For these signals, the phase is discontinuous with frequency. If the phase $\emptyset \left(f\right)$ is disregarded in Equation (3), the so-called “real cepstrum ${Ceps}_{r}\left(\tau \right)$” can be obtained as follows:(4)$${Ceps}_{r}\left(\tau \right)=IFFT(\log A\left(f\right)).$$

The real cepstrum has the advantage that it can be applied to both forcing and response signals, as it does not require phase to be a continuous function of frequency. However, neglecting the phase means that the cepstrum is only reversible to the spectrum domain, not back to the time domain. It can also be used to smooth auto spectra, which will usually reduce noise. One key property of the cepstrum domain is that the forcing function and transfer function effects become additive in the cepstrum, thus enabling their separation. For single input multiple output (SIMO) systems, any output signal $r\left(t\right)$ is the convolution of the input signal $x\left(t\right)$ (forcing function) with the impulse response function of the transmission path g(t) (transfer function), as given in Equation (5). The convolution (∗) is converted to a multiplication in the frequency domain (Equation (6)) and to an addition process (Equation (7)) by the log function in the cepstrum operations as follows [35]:(5)$$r\left(t\right)=x\left(t\right)*g(t)$$
(6)$$R\left(f\right)=X\left(f\right)\cdot G(f)$$
(7)$$\log\left(R\left(f\right)\right)=\log\left(X\left(f\right)\right)+\log\left(G\left(f\right)\right).$$

More comprehensive information and advice on the use of the cepstrum can be found in [36]. Cepstrum analysis has been widely used in a number of applications, e.g., radar signal and marine exploration [34], biomedical signal processing [37], speech analysis [38] and mechanics [35]. In speech analysis, the cepstrum is used to determine the voice pitch (performed by measuring the harmonic spacing) and to separate the formants, i.e., the transfer function of the vocal tract, from voiced and unvoiced sources [38]. Cepstrum analysis for gearbox health monitoring was initially proposed in [39] through the realization that the cepstrum does not only detect families of harmonics, but also equally spaced modulation sidebands. In this sense, the cepstrum serves as an effective tool to characterize families of harmonics, sidebands and modulations in a clear and easy-to-interpret format, as these families become concentrated in rahmonics. This makes the cepstrum an important tool for gear health monitoring, as local faults in gears give an impulsive modulation of the gear mesh signals (both amplitude and frequency modulation) that results in sidebands spaced at the speed of the gear on which the local fault is located [35]. A number of researchers have made use of the cepstrum to detect and isolate the source of a local gear fault to a particular gear set, e.g., [40,41,42].

Figure 9 presents the cepstrum domain plot for the Ring Front sensor record #522. This is illustrated using four subplots to highlight families of rahmonics across different quefrency ranges. Interesting observations can be seen here that are not seen in both the time and frequency domains of Figure 5 and Figure 6, respectively. Figure 9a displays the low quefrency region (0.001 to 0.5 planet rotations), where transfer path effects (transient low quefrency part) can be seen clearly in addition to a gear mesh rahmonic at 0.0286 (i.e., 1/35). Figure 9b shows the quefrency range from 0.5 to 45 planet rotations, where two families of rahmonics are evident. The first rahmonic family relates to the planet carrier and occurs at multiples of 2.82857 planet rotations (i.e., 99/35); the second family relates to the planet rotation and occurs at integer multiples of the planet rotation. High amplitudes of the rahmonic family related to the planet gear rotation are seen at exactly 17 and 34 planet rotations, which are very close to adjacent rahmonic components of the planet carrier. This is further shown in Figure 9c,d, which provide zoomed-in views around the 6th rahmonic of the carrier and the 17th rahmonic of the planet, and the 12th rahmonic of the carrier and the 34th rahmonic of the planet, respectively.

##### Cepstrum-Editing Technique

Sawalhi and Randall [23] proposed a cepstrum-editing algorithm to remove selected families of harmonics and sidebands by editing the amplitudes (setting families of rahmonics to zero) in the real cepstrum, transforming back into the logarithmic domain, where the initial phase was restored, and then transforming the signal back to the time domain. A flowchart illustrating the cepstrum-editing process is presented in Figure 10. The process has been successfully used to diagnose faults in bearings [31,43], and it has been recently used as a gear diagnostic method denoted as the “angle cepstrum comb liftering (ACCL) method” [24]. In [24], the authors used comb liftering in the cepstrum of the order tracked acceleration signal to select all rahmonics of the gear mesh quefrency of a particular meshing gear pair. This was followed by the application of a notch filter to remove gear mesh frequencies to enhance the detection of gear faults. This method was found to be effective for local fault detection and trending, such as tooth root cracks and localized spalls.

##### Proposed Cepstrum-Editing Scheme

For the planet rim crack propagation dataset, a proposed cepstrum-editing scheme for fault detection in planet gears is presented in the schematic diagram of Figure 11. The overall process uses the structure shown in Figure 10 along with insights from recent cepstrum editing for gears presented in [24]. The method works by removing (or setting to zero) all quefrency components except the DC (retained for scaling) and several integer quefrency lines around each planet rahmonic. This is equivalent to the use of a comb filter with a certain tine width (*Tw*) in the cepstrum domain. Through this cepstrum-editing scheme, the effect of the transfer function (low quefrency part) is removed, and thus, the spectrum becomes white. Gear mesh harmonics are also eliminated, and, as such, there is no need to use a notch filter to remove these harmonics as was carried out in [24].

Note that the effect of Tw in the cepstrum domain corresponds to multiplying the cepstrum with a periodic rectangular function with equivalent width of Tw lines, spaced by an amount corresponding to the number of lines (samples) per planet period [24]. In the log spectrum domain, this corresponds to a convolution with a *sinc*(*x*) function with the first zero crossing at *Fs*/Tw shaft orders, where *Fs* is the order sampling frequency (samples/revolution) [24]. An appropriate tine width (Tw) can be judged with respect to the equivalent frequency band of the discrete spectral components protruding from the base noise level [24]. In order to select an appropriate Tw, a first approach was to define a low-pass (*LP*) filter cut-off frequency. This was selected as 2.5× GMF (i.e., at 87.5× planet shaft orders). This gives an initial guide for a possible choice of Tw by ensuring harmonics do not decrease considerably at the low-pass cut-off frequency. To do so, the zero crossing of the *sinc*(*x*) function was set arbitrarily at 2 times the required *LP* cut-off frequency, i.e., at 5× GMF. This corresponds approximately to 175 planet shaft orders, i.e., 35 × 5. At a sampling frequency *Fs* of 4095 samples/revolution, this corresponds to using a tine width (Tw) of 23 lines (4095/175), i.e., (*Rp* ± 11) where *Rp* is the central planet rahmonic. A second approach was based on minimizing the squared error between two consecutive tine widths Tw1 and Tw2 in similar terms with the MED filter length selection criterion reported in [44]. If $d_{i\left(Tw1\right)}$ and $d_{i\left(Tw2\right)}$ represent the column vectors of the low-pass cepstrum filtered outputs at 2.5× GMF using Tw1 and Tw2,respectively, a squared error can be calculated using the following:(8)$$E\left(Tw1\right)=\sum_{i=1}^{N}\left.\left\{{\left.{\left[d_{i\left(Tw1\right)}-\right.d}_{i\left(Tw2\right)}\right]}^{T}\left[d_{i\left(Tw1\right)}\right.\left.{-d}_{i\left(Tw2\right)}\right]\right.\right\}.$$

An appropriate tine width, Tw, would be one corresponding to [$E\left(Tw\right)$]_min_, as a variation in output signal would be least when centred on the appropriate/optimum tine width. The limits of the search were chosen between Tw = 9 (*sinc*(*x*) zero crossing of 13× GMF) and Tw = 33 (*sinc*(*x*) zero crossing of 3.5× GMF). The result of this search is plotted in Figure 12, with a minimum squared error at Tw=25, i.e., (*Rp* ± 12) and kurtosis values of just over 22. Note that this optimum Tw of 25 is close to the first approach selection of 23, which was based on setting the zero crossing of the *sinc*(*x*) function at 2 times the *LP* filter cut-off frequency. The power spectral density results of using a *Tw* of 25 are presented in Figure 13 (scaled in dB). The effect of using a rectangle window of Tw=25 can be seen clearly in the spectrum of the filtered cepstrum in Figure 13b, where levels are noticed to decrease considerably at approximately 165 planet orders (approximately 4.7× GMF), with relatively constant amplitudes within the *LP* filter limits. This is further demonstrated in Figure 14 by plotting *sinc*(*x*) function with a Tw=25 (red) and overlaying it on the edited cepstrum (blue) and its envelope (black).

#### 2.3.2. Blind Deconvolution (BD)

Blind deconvolution is one of the radical and commonly used sparse feature extraction/enhancement methods in fault diagnosis. Its fundamental working principle relies on finding an inverse filter using a measured signal to extract/enhance fault-related features (impulses) through maximizing an appropriate objective function. There are two main methods to find the inverse filter, which are the objective function method (OFM) and the Eigen vector algorithm (EVA). Chen et al. [45] provided a summary of these two methods for rolling element bearing feature extraction and proposed the use of a squared envelope spectrum as the objective function. EVA methods have been widely used recently as they are non-iterative and do not require the objective function to be differentiable, as is the case with OFM. The minimum entropy deconvolution (MED) technique [25] is one of the earliest OFM-based methods which uses kurtosis (impulsiveness measure) as its objective function. MED was used effectively to enhance the impulses arising from spalls and cracks in gears [32] and in enhancing spall detection in rolling element bearings [46]. One of the shortcomings of MED is that it tends to enhance single transients (random impulsive noise) and, as such, may result in wrong conclusions. A second shortcoming of MED is its reliance on kurtosis, which is a sensitive measure to random transients or outliers in the signal. To overcome these shortcomings, a number of BD algorithms have been proposed. These methods were either frequency-independent or frequency-dependent in a quest to target/enhance only periodic impulses related to faults in rotating machinery. Examples of frequency-independent approaches are the optimal MED [47], the maximum Gini index deconvolution [48,49] and blind filtering based on Box-Cox sparse measures [50]. Some of the frequency-dependent algorithms are maximum correlated kurtosis deconvolution (MCKD) [51], multipoint optimal minimum entropy deconvolution adjusted (MOMEDA) [52], sparse maximum harmonics-to-noise-ratio deconvolution (SMHD) [53] and maximum second-order cyclostationarity blind deconvolution (CYCBD) [54].

In this work, MED is selected as a means to enhance the impulsiveness of the output of the cepstrum-edited low-pass signal. MED is a fast and robust algorithm when applied to signals with no random transients/outliers. The use of the proposed cepstrum-editing technique ensures that features related only to the planet rotation are extracted. This, along with low-pass filtering, ensures that random impulses are eliminated. The use of kurtosis as an indicator is also desirable as the aim is to enhance signal impulsiveness. There is still, however, a possibility of having a one-off high impulse occurrence in the H-SSA signal related to planet rotation, from a shock or measurement-related error, which will result in high kurtosis levels in some records. These odd records can be dealt with by using a median smoothening filter during trending, as discussed later in Section 3.2.

##### Minimum Entropy Deconvolution (MED)

MED [25] is a system identification method used originally to extract reflectivity information from seismic data to identify and locate layers of subterranean minerals. MED has shown its effectiveness in de-convolving impulse excitations from a mixture of response signals [55,56]. An illustration of the blind convolution concept is shown in Figure 15 [57]. In this illustration, it is assumed that an observed fault signal *d*(*t*) (in this case, the output of cepstrum editing after being low-pass filtered) is a convolution of some transfer function *w*(*t*) (e.g., transfer path effect) with the fault impulse series *r*(*t*) plus added superimposed noise *n*(*t*). The goal of the blind deconvolution is to recover the fault impulse series *r*(*t*) by removing/reducing the effect of the transfer function. In other words, the aim is to find an inverse filter (*g*) which creates *y*(*t*), where *y*(*t*) *= g*
∗
*d*(*t*) such that *y*(*t*) is a representation of *r*(*t*), which could be a shifted and scaled version of *r*(*t*). MED works by searching for an optimum set of filter coefficients (de-convolving filter) to recover the output signal of an inverse filter with the maximum value of kurtosis, where kurtosis is a fourth-order statistic that measures the deviation from Gaussianity in the signal.

Wiggins [25] proposed to solve the deconvolution by maximizing the kurtosis of the deconvolution output *y*(*t*) defined for a digitized signal *y* with *N* samples using Equations (9) and (10):(9)$$Kurtosis=\frac{\frac{1}{N}\sum_{i=1}^{N}{\left(y\left(i\right)-\bar{y}\right)}^{4}}{\sigma^{4}}$$
(10)$$Standard\;deviation\;(\sigma )=\sqrt{\frac{1}{N}\sum_{i=1}^{N}{\left(y\left(i\right)-\bar{y}\right)}^{2}}$$

The solution can be achieved iteratively using the objective function method (OFM) [55]. OFM is an optimization process designed to maximise the kurtosis (or minimise the entropy) of the MED output. This is achieved by changing the values of the coefficients in the MED filter (*g*). The sequence of the process consists of the following:Assume initial values of the filter, *g = f*
^(1)^
*=* [*g*_0_, *…*, *g_L_*]*^T^* where *L* denotes the length of the filter.Calculate vector *f*
^(1)^: New filter coefficient defined as the correlation between *y*^3^(*t*) and the data *d(t)*. *f*
^(1)^ is found by solving the system *Rg = f*, where *R* is an *N* × *N* autocorrelation matrix defined with the *N* as the first delay of the autocorrelation function of the data *d*(*t*).Calculate the error (*err*) using Equations (11) and (12)
(11)$$err=\left(f^{\left(1\right)}-\mu f^{\left(2\right)}\right)/\mu f^{\left(1\right)}$$
(12)$$\mu ={({{E(f}^{\left(1\right)})}^{2}{{E(f}^{\left(2\right)})}^{2})\;}^{1/2}$$

The optimization process finishes when the values of the coefficients converge within the specified tolerance (*err* < 0.001). If *err* > 0.001, then filter coefficient $f^{\left(2\right)}$ becomes $f^{\left(1\right)}$, and the process is repeated. A condition is also set to abort the iteration if the value of *err* is found to be diverging.

The choice of the filter length is arbitrary. However, the longer the filter is, the slower the calculations become with no clear benefits on signal clarity and fault detection. MED de-convolved output is highly dependent on the choice of *L*, which mainly is a function of the transfer function effect and on the position of the spikes as well. Excessively long filter lengths may oversimplify the output, while filter lengths that are too short may not resolve the output spikes. A simple criterion that was reported to give good results is the so-called ‘simplified MED operator length criterion’ [44]. This criterion is based on minimizing the squared error between two filter lengths *L* and *L* + 1. If *y_i_*(*L*) and *y_i_*(*L* + 1) are the column vectors of the MED outputs for two filter lengths *L* and *L* + 1, a squared error *E*(*L*) can be defined as follows:(13)$$E\left(L\right)=\sum_{i=1}^{N}\left.\left\{{\left.{\left[y_{i\left(L\right)}-\right.y}_{i\left(L+1\right)}\right]}^{T}\left[y_{i\left(L\right)}\right.\left.{-y}_{i\left(L+1\right)}\right]\right.\right\}$$

A possible good length *L* would be one that corresponds to the minimum value of *E*(*L*), where changes in the de-convolved output for perturbations in the filter length will be least when centred around the correct filter length [44]. A demonstration of the use of this criterion is presented in Figure 16 for filter lengths of 10 to 1500. It can be seen from Figure 16b that a number of filter lengths (local minima) give good results with the lowest one at 26. The optimum filter length within the search boundaries was found to be 945, which is approximately one-quarter of the planet rotation or 8 tooth mesh periods.

## 3. Results

### 3.1. Cepstrum Editing and MED Filtration of a Typical Result (Record #522)

Figure 17 provides a typical result of crack feature extraction and enhancement using the new cepstrum-editing method with and without MED filtering. The result presented is taken from record #522 for the Ring Front sensor and presents 35 planet gear rotations (out of the 99 rotations), i.e., just above 12 carrier rotations. The raw signal Figure 17a has a low kurtosis value of 3.26 and is dominated by gear mesh frequencies. Cepstrum editing increased the kurtosis levels to 6.08, as shown in Figure 17b. The cepstrum-edited signal with low-pass filter at 2.5× GMF is plotted in Figure 17c. Impulses (planet-ring mesh) can now be seen clearly at every revolution of the planet gear, with smaller ones (planet-sun mesh) seen in between. Figure 17d shows the outcome of enhancing the cepstrum-edited low-pass filtered signal using MED, where kurtosis levels increased notably from 22.28 to 36.11 indicating a clearer impulse enhancement and reduced noise level. The use of MED provides a much clearer picture of impacts at every rotation of the planet gear, as can be seen from Figure 17d where high amplitudes are seen at planet rotations 3, 6, 9, 10, 13, 14, 16 and 17. The highest impacts are observed to repeat every 17-planet rotation, as was seen earlier from the cepstrum analysis provided in Figure 9. Note that 17 revolutions of the planet is very close to 6 (i.e., 17 × 35/99 = 6.01) revolutions of the carrier—approximately 1% off being an integer number—so the relative position of the planet and the carrier would almost be the same every 17 planet rotations and thus be reinforced in the averaging process. Results obtained from other sensors are plotted in Figure 18. This shows a clear detection of impulses with higher kurtosis values compared to the synchronous averaged signal results presented in Figure 7.

### 3.2. Trending Results

Figure 19 provides a comparison of rim crack trending using kurtosis values of residual signals obtained by the traditional approach, cepstrum editing with LP (no MED) and cepstrum editing enhanced with LP and MED. The trends have been calculated using rolling median averaging to suppress outliers, where 9 samples were used for the traditional approach, 12 samples were used for cepstrum editing and 20 samples were used for cepstrum editing with MED. The use of different numbers of samples for the rolling median averaging in each method comes as a result of the increased effectiveness of recovering more transient information in each successive processing step (as evidenced by the signals progressively higher kurtosis values). Longer median filters are needed to suppress those records that have the odd one-off transients. It can be seen from Figure 19 that cepstrum editing provides an earlier detection (detection defined as two sensors with kurtosis levels of 3.5 and above) at record #247 compared to record #260 using the traditional approach. Cepstrum editing also provides a better trending capacity through an upward trending starting at record #319 compared to late upward trending at record #450 in the traditional approach. The enhancement of cepstrum editing using MED is further obvious through an early detection at record #144 and an upward trending starting at record #263.

Residual signals of the earliest detection records are plotted in Figure 20, Figure 21 and Figure 22 for the traditional approach, cepstrum editing, and cepstrum editing with MED, respectively. Note that Figure 21 and Figure 22 are plots of the squared envelope residuals obtained from the cepstrum processing that have been reshaped into a [99 × 1] array and plotted using a 2D surface plot in order to better show the recurring amplitude pattern generated by the fault, as discussed further in Section 4.

Trending using scaled kurtosis [58] (as seen in Figure 23) provides a further clear picture about the improved results obtained using cepstrum editing and the even better results when using MED. The scaled kurtosis is obtained by scaling the residual kurtosis by the ratio of the residual signal standard deviation, or root mean square (RMS), to the residual signal RMS of the first (i.e., healthy) measurement. It was proposed to account for the changes at the beginning of fault development and later stages, where early stages of the fault have high spikiness while later progression stages see increased energy levels [58]. The scaled kurtosis trend of the cepstrum editing enhanced with MED in Figure 23c shows clearly three stages of crack growth which can be denoted by the initial growth (from record #1 to Record #350); the sustained growth (from record #351 to Record #450) and the accelerated growth (from record #451 to Record #526).

### 3.3. Clustering Results

A traditional machine learning clustering technique (Kmeans++ [59]) has been employed to identify the three different stages of the crack growth, i.e., the initial growth, sustained growth and accelerated growth, through mapping the three most sensitive channels up to measurement number 500. The K-means clustering method is a widely used technique that aims at minimizing the average squared distance between points in the same cluster [39]. Kmeans++ is a means of improving the K-means technique through improving the speed and accuracy by using a randomized seeding technique. K-means partition the points in the (*n* × *p*) matrix into *k* clusters by minimizing the sum of squared distance between each point and the closest centroid, where *k* is the number of centroids/clusters and *n* × *p* is data points (trending parameters, where *n* is rows and *p* is variable).

To examine the clustering quality, the Silhouette plot is used. The Silhouette plot provides a measure of how close each point in one cluster is to points in the neighbouring cluster, and as such, offers a means to assess clustering parameters like the number of clusters visually. The higher the average Silhouette score, the better clustering it achieves. Silhouette coefficients have a range of [–1, 1], where 1 means a point is far away from neighbouring cluster, 0 indicates a point is on or close to neighbouring cluster and a *negative value* designates a point might have been assigned to wrong cluster. Figure 24 presents an assessment of K-means++ clustering using Silhouette coefficient values for the traditional, cepstrum editing and the cepstrum editing enhanced with MED approaches. Cepstrum editing enhanced with MED achieved the highest average Silhouette score (best clustering) of 0.75 compared to 0.73 for cepstrum editing and 0.67 for the traditional approach. A further clear improvement achieved can be seen in the separation of cluster 2 (sustained growth) from cluster 3 (accelerated growth), where only one record (#455) was below 0 for the cepstrum editing with MED, indicating better separation compared to the traditional approach. Figure 25 provides a 3D scatter presentation of the three clusters, showing an improved clustering when using cepstrum editing, Figure 25b, compared to traditional, Figure 25a, and the further improvement obtained when enhancing cepstrum editing with the use of MED, Figure 25c.

## 4. Discussion

Cepstrum analysis provides informative insights into signals, not often seen in both time and frequency domains. This is because it separates the forcing and transfer functions and provides an easier interpretation to families of harmonics and sidebands. The example presented in Figure 9 clearly supports the benefits of using cepstrum analysis routinely in gearbox health monitoring programs. Figure 9 shows clear periodicity at the carrier frequency and the planet rotation frequency and a very clear periodicity at every 17th rotation of the planet gear, which are not seen in both the time and frequency domain nor in the residual of the P-SSA signals when using the traditional analysis tool. Although the traditional technique using P-SSA residuals gives a clear diagnosis of the fault and provides a good trending capability, it falls short of the proposed cepstrum-editing method enhanced with MED. The latter provides a very clear detection of the fault at a much earlier stage, it has a better trending and clustering capacity and gives further insights into the impact amplitudes at every rotation of the planet, as can be seen in Figure 17d.

In order to gain more insight into the impact amplitude variation seen in Figure 17d and through cepstrum analysis, the squared envelope and its spectrum of the Ring Front sensor (signal presented in Figure 17d) were produced and are plotted in Figure 26. The squared envelope shows different amplitudes at each planet-ring mesh impact due to the variation of the impact locations with respect to the fixed sensor location. The spectrum shows the harmonics of 1× planet gear rotation with high amplitudes at 2× due to the presence of two impact events every planet rotation (planet-ring and plant-sun meshes). Modulation sidebands are seen at 1× carrier frequency (i.e., 0.3535× planet) and at 1.06061× carrier (i.e., 0.29293× planet) as well as at the difference frequency of 0.06061 (corresponds to 6 planet teeth difference).

Modulations can be understood by realising that each planet gear completes one rotation every 360 × 35/99 of the carrier rotation, i.e., every 127.2727° of carrier rotation or at 0.3535× carrier rotation. If a fixed sensor is placed at 0°, the first planet-ring meshing spike will take place after the planet gear moves 127.2727°, while the third impact will be after the carrier completes 1.06061 rotations. This will be a large impact compared to 1 and 2, as it will have a phase difference of 21.8°.

Table 3 lists planet locations with respect to a ring sensor located at 0°, where the amplitude seen in the squared envelope is in agreement with predicted locations. For example, impact number 12 has a low amplitude, as it occurs almost 90° out of phase. Impacts at every 17 rotations have the highest amplitudes, as the rim crack will approximately align with the sensor location (minimum phase difference of 3.6 degrees corresponding to one tooth difference). Note that impact strengths seen in the normalized squared envelope after each planet rotation agrees well with the predictions provided in Table 3.

## 5. Conclusions

This paper presented a signal processing algorithm to detect, enhance and trend rim cracks in planet gears of helicopter main transmission gearboxes. The algorithm is based on using a cepstrum-editing technique to extract harmonics and sidebands of the planet gear from the hunting-tooth synchronous averaged signals and low-pass filtering and minimum entropy deconvolution (MED) to enhance the fault features. The algorithm has been successfully tested on a vibration dataset collected from a planet gear rim crack propagation test undertaken in the Helicopter Transmission Test Facility (HTTF) at DSTG Melbourne. The presented algorithm is fundamentally built on angular resampling, Fourier transform and inverse Fourier transform (for cepstrum), autocorrelation (MED) and low-pass filtering. Therefore, the algorithm does not require high intensity computations and, as such, can be easily deployed in real-world applications. The integration of the proposed algorithm into maintenance strategies for helicopter planetary gear systems can be straightforward by combining the new condition indicators (CIs) generated by the algorithm with other indicators in an existing HUMS/PHM system. The trending capability of the proposed algorithm is specifically important to be integrated into maintenance strategies for helicopter planetary gear systems. Furthermore, if there is a deep machine learning component in the predictive maintenance framework, the new CIs can be used as input for the neural network to make critical decisions on issuing warnings to operators in conjunction with the trending capability of the proposed algorithm.

This paper discussed a number of parameter choices in relation to the algorithm, the first being the tine width of the comb filter used in the cepstrum editing. The tine width can be deduced by setting the *sinc*(*x*) zero crossing to two times the low-pass filter length or by minimizing the squared error between two consecutive tine widths and selecting the minimum error. For a low-pass filter at 2.5× GMF, a tine width of 23 or 25 was found appropriate. The second choice relates to the inverse filter length of MED. It has been noted that this choice can be arbitrary, but a simple guide is to minimize the squared error between different filter lengths and select the one with the minimum error. A filter length of 945 was found suitable when searching between 10 and 1500 filter lengths. The use of MED as an enhancement tool assumes no random outliers and that cepstrum editing did a good job in eliminating these. In cases where some random outliers still appear at any planet rotations, smoothening with median filtration during the trending process has been used to remove this effect.

Although the traditional technique using P-SSA residuals gives a reasonable diagnosis of the fault and provides a good trending capability, it falls short of the proposed cepstrum-editing method enhanced with MED. The use of the cepstrum-editing technique provides a much earlier detection capability compared to the traditional approach. When cepstrum editing is enhanced with MED, the first crack detection is achieved at an even earlier stage, indicating a superior capability of detecting incipient faults. This is also evident from the achievement of better trending and clustering results. Cepstrum analysis provides further informative insights into the signals, often not seen in the time and frequency domains. This is because cepstrum analysis separates the forcing and transfer functions and provides an easier interpretation to the families of harmonics and sidebands. The squared envelope signal has shown different amplitudes at each planet-ring impact due to the variation of the planet location with respect to the fixed sensor location.

## Figures and Tables

**Figure 1 sensors-24-02593-f001:**
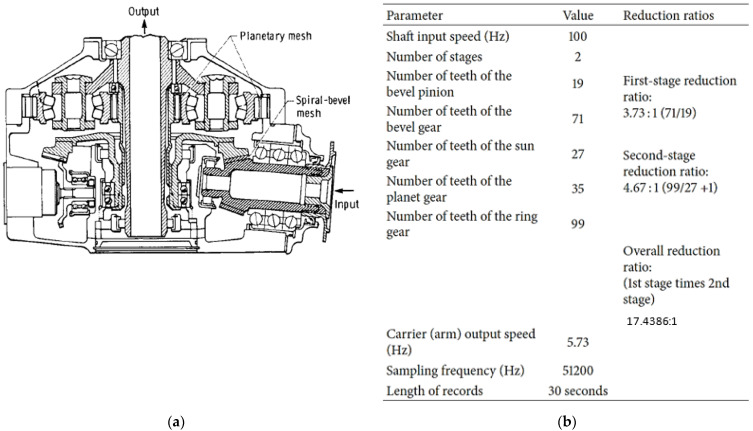
(**a**) Bell 206 3-planet schematic presentation [27] (**b**) key information and frequencies.

**Figure 2 sensors-24-02593-f002:**
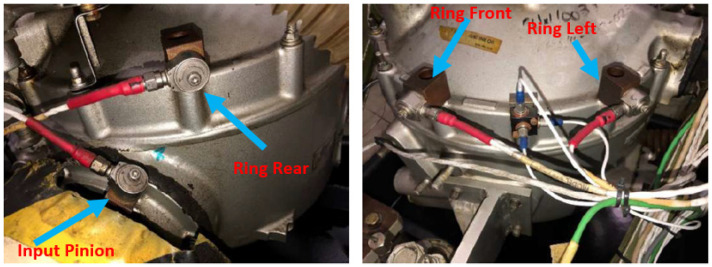
Physical locations of HTTF accelerometers.

**Figure 3 sensors-24-02593-f003:**
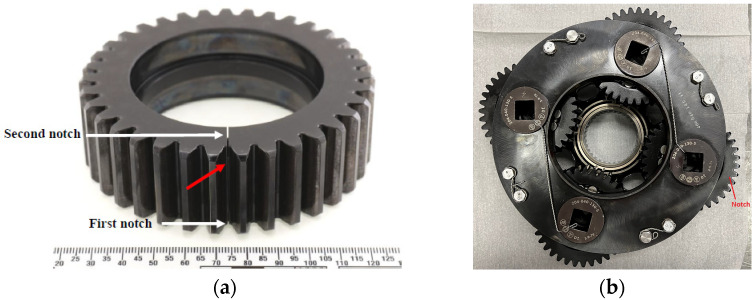
Four-planet setup and the physical location of the gear notches (**a**) the notched planet gear prior to the second part of the test; (**b**) bottom view of the planetary assembly.

**Figure 5 sensors-24-02593-f005:**
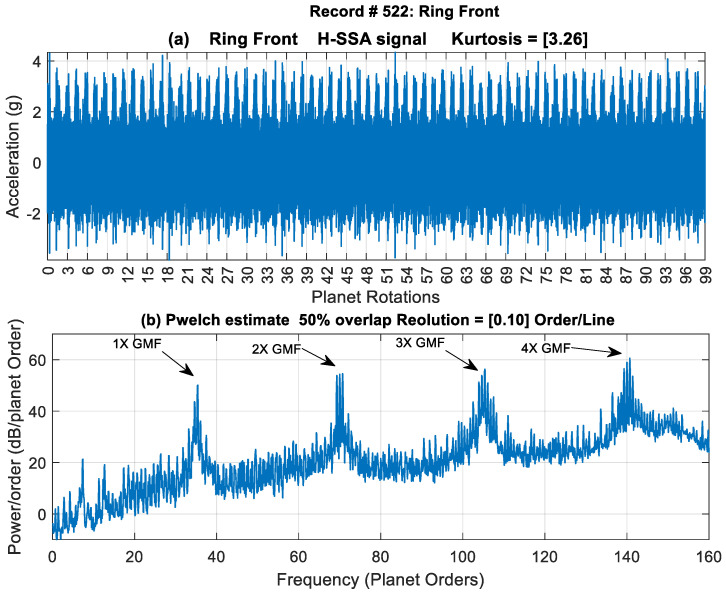
Record #522 time and frequency signatures.

**Figure 6 sensors-24-02593-f006:**
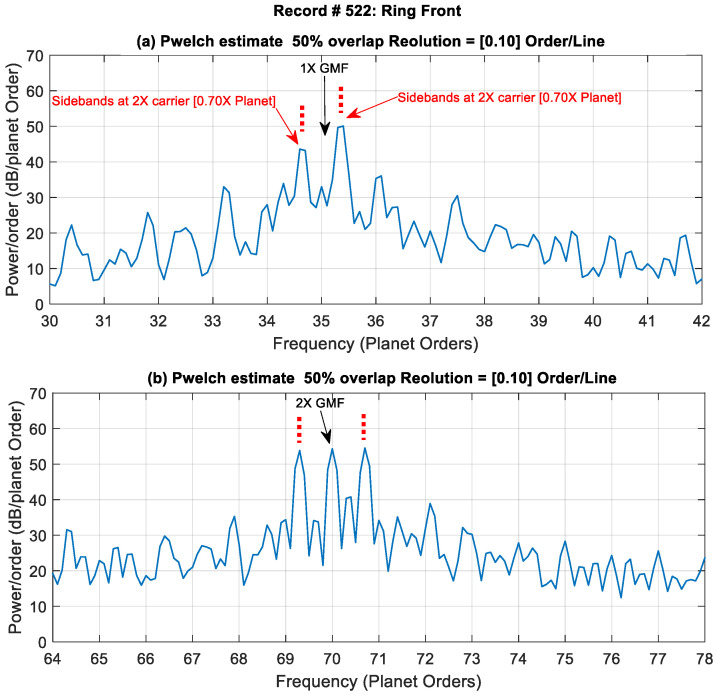
A zoomed-in plot showing sidebands around the 1st and 2nd gear mesh frequencies.

**Figure 7 sensors-24-02593-f007:**
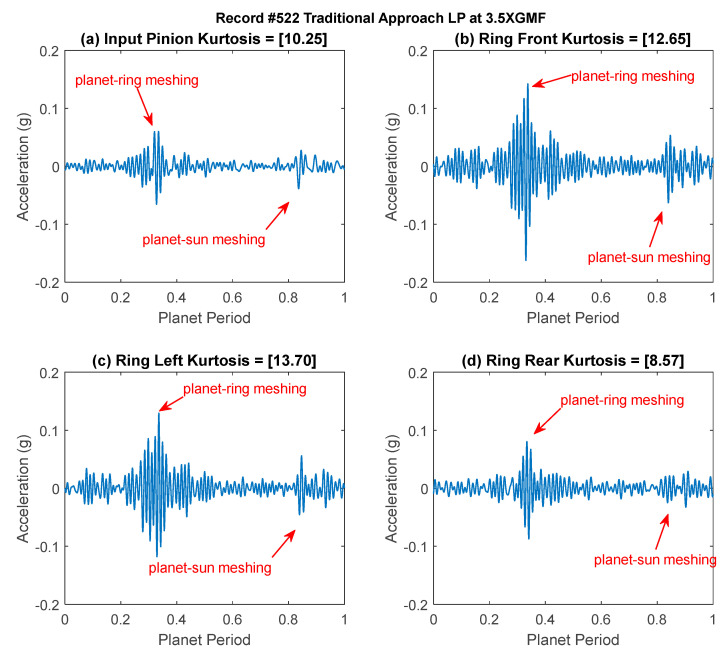
Residual signals of record #522 (**a**) Input Pinion (**b**) Ring Front (**c**) Ring Left (**d**) Ring Rear.

**Figure 8 sensors-24-02593-f008:**
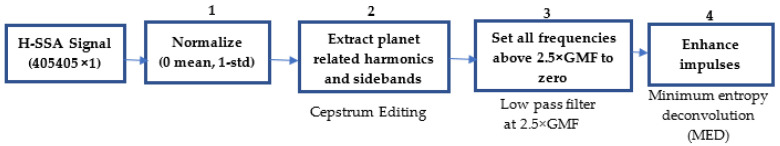
Signal processing algorithm to extract and enhance rim crack impulses.

**Figure 9 sensors-24-02593-f009:**
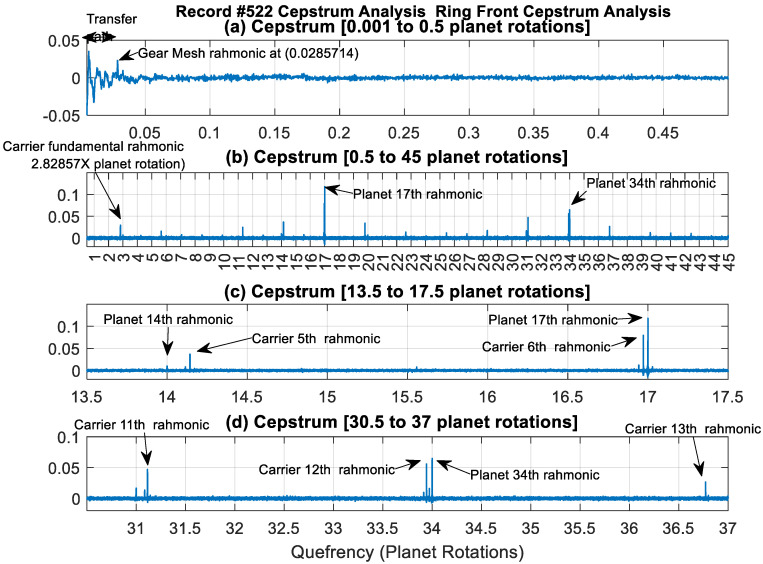
Cepstrum content of Ring Front sensor at record #522.

**Figure 10 sensors-24-02593-f010:**
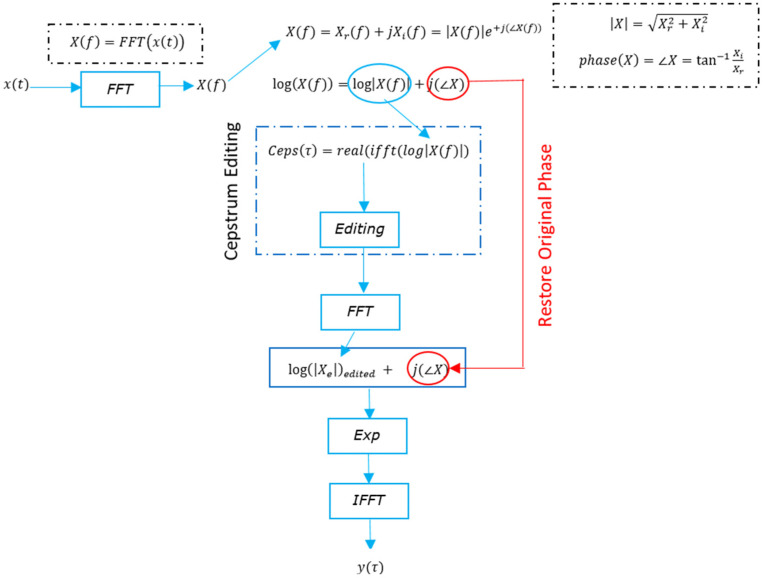
The flowchart for the cepstrum-editing technique.

**Figure 11 sensors-24-02593-f011:**
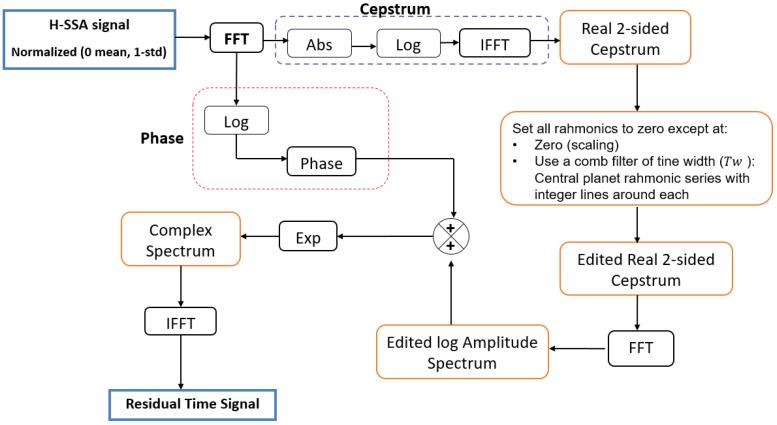
The flowchart for the new cepstrum-editing scheme.

**Figure 12 sensors-24-02593-f012:**
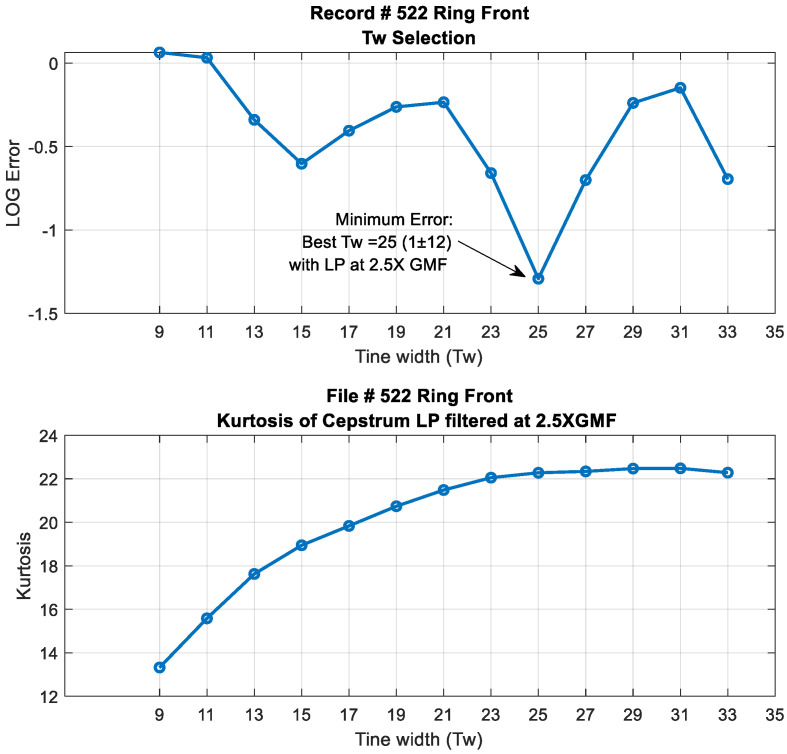
Tine width selection criterion by minimizing the squared error between consecutive low pass cepstrum filtered outputs. Top: Log error vs. Tw. bottom kurtosis values vs. Tw.

**Figure 13 sensors-24-02593-f013:**
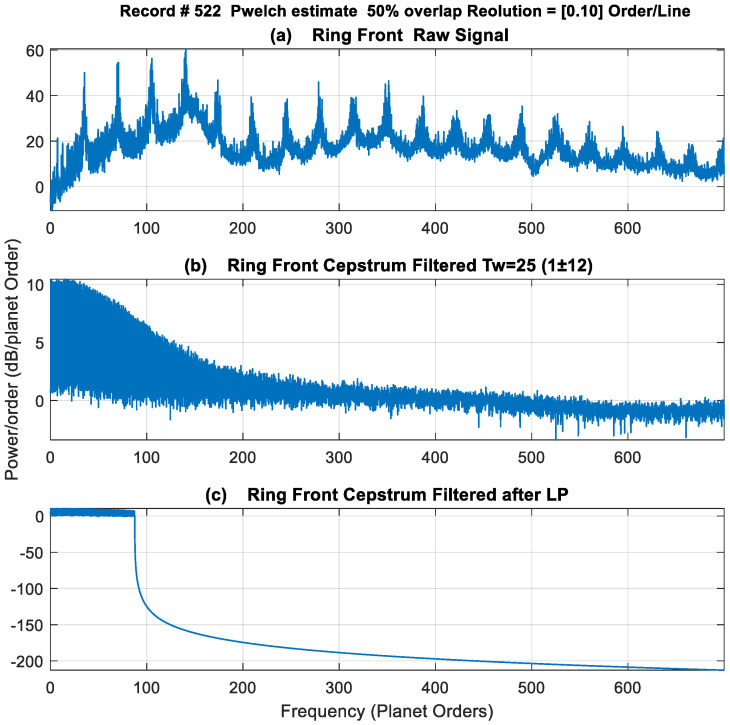
Power spectral density (dB re 1 × 10^−6^) (**a**) Ring Front raw signal (**b**) cepstrum filtered with TW = 25 (**c**) low-pass filter at 2.5× GMF.

**Figure 14 sensors-24-02593-f014:**
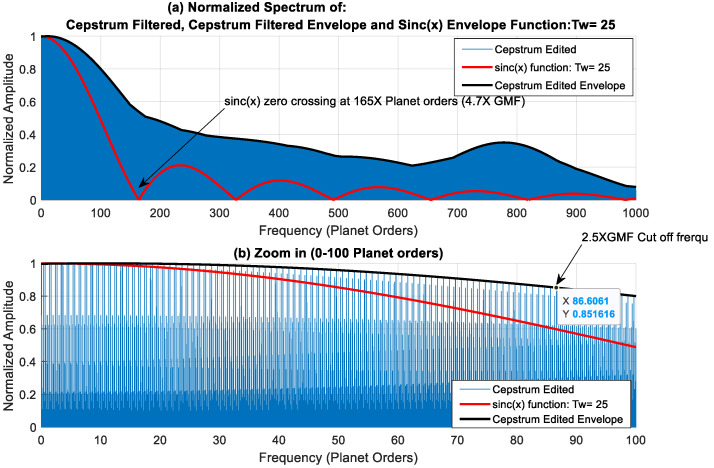
Normalized spectrum of cepstrum filtered signal, cepstrum filtered envelope signal and *sinc*(*x*) envelope (absolute value) function: Tw = 25 (**a**) 0–1000 planet orders (**b**) 0–100 planet orders.

**Figure 15 sensors-24-02593-f015:**
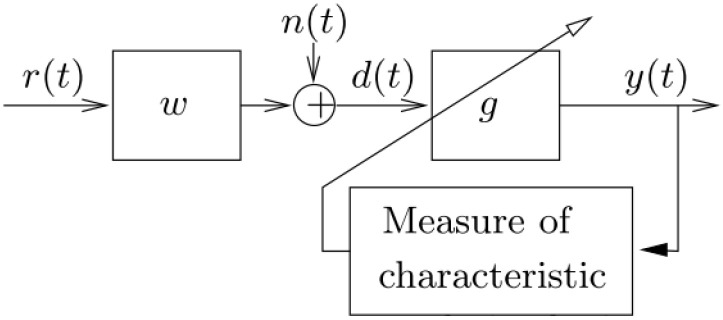
Illustration of blind deconvolution.

**Figure 16 sensors-24-02593-f016:**
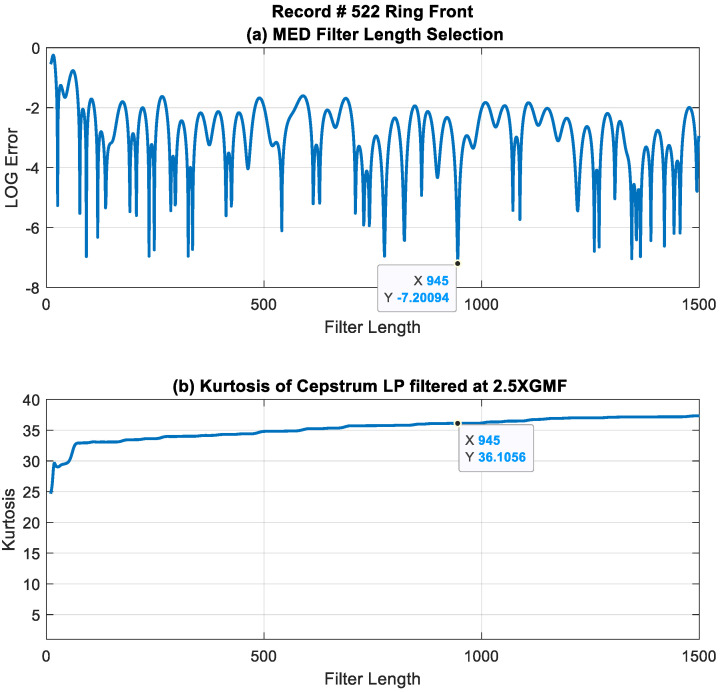
(**a**) MED filter length selection using minimum error (**b**) kurtosis of the output signal (cepstrum LP enhanced with MED).

**Figure 17 sensors-24-02593-f017:**
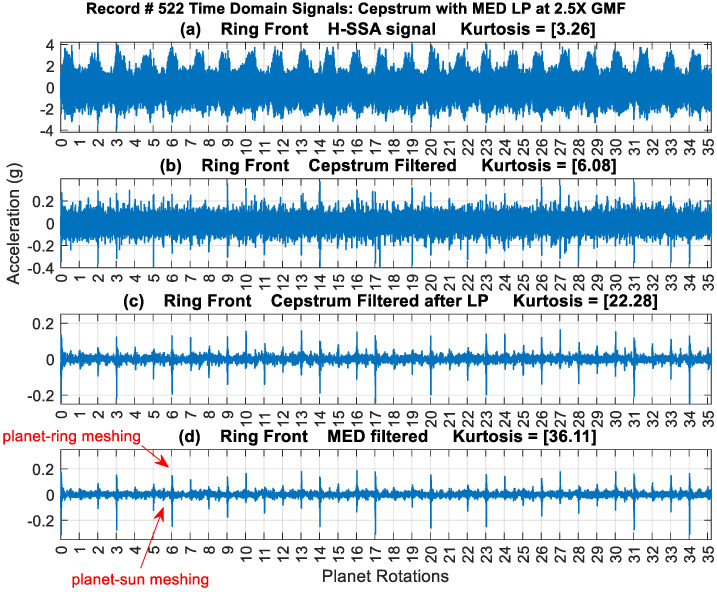
A total of 35 planet rotations of Ring Front sensor signal of record #522 (**a**) raw signal (**b**) cepstrum filtered (**c**) cepstrum filtered after low-pass filtering (2.5X GMF) (**d**) signal c after MED filtration.

**Figure 18 sensors-24-02593-f018:**
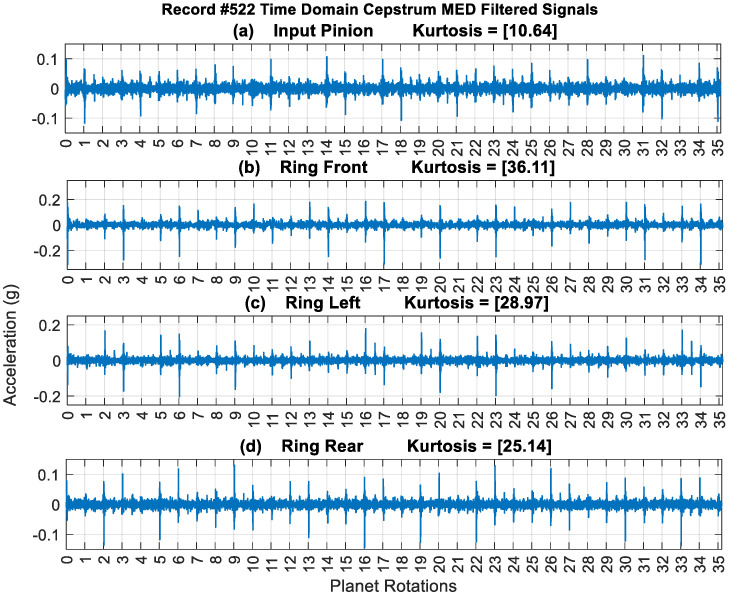
A total of 35 planet rotations: cepstrum-edited low-passed and MED filtered of the four sensors, record #522 (**a**) Input Pinion (**b**) Ring Left (**c**) Ring Front (**d**) Ring Rear.

**Figure 19 sensors-24-02593-f019:**
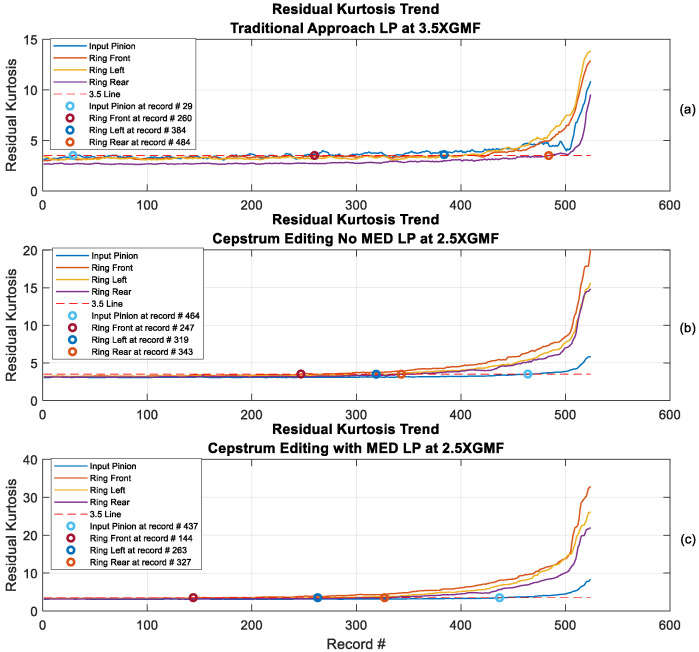
Residual kurtosis trending: (**a**) Traditional approach with low-pass (LP) filter at 3.5× GMF (rolling median nine samples), (**b**) Cepstrum editing with no MED (LP at 2.5 GMF: rolling median 12 samples), (**c**) Cepstrum editing with MED (LP at 2.5 GMF: rolling median 20 samples).

**Figure 20 sensors-24-02593-f020:**
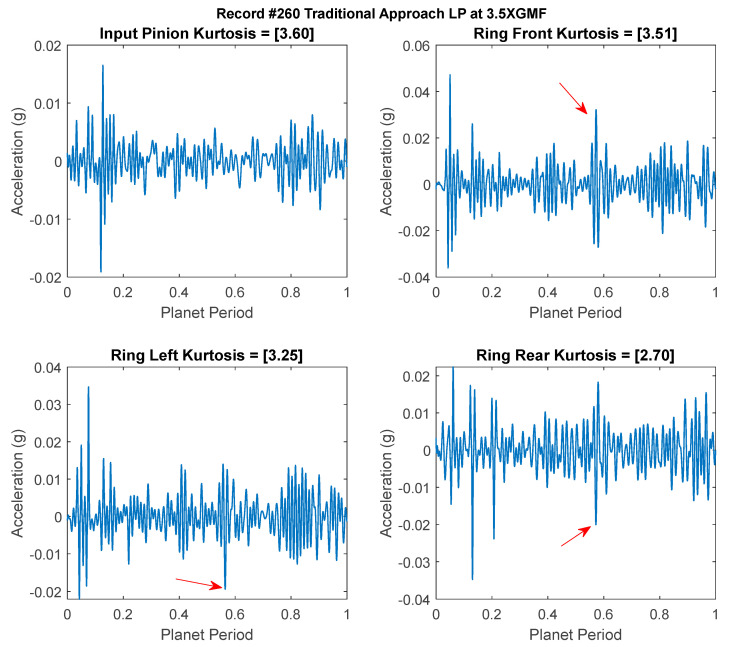
Traditional approach earliest crack detection at record #260 (the red arrows indicates the impacts likely induced by planet-sun gear mesh).

**Figure 21 sensors-24-02593-f021:**
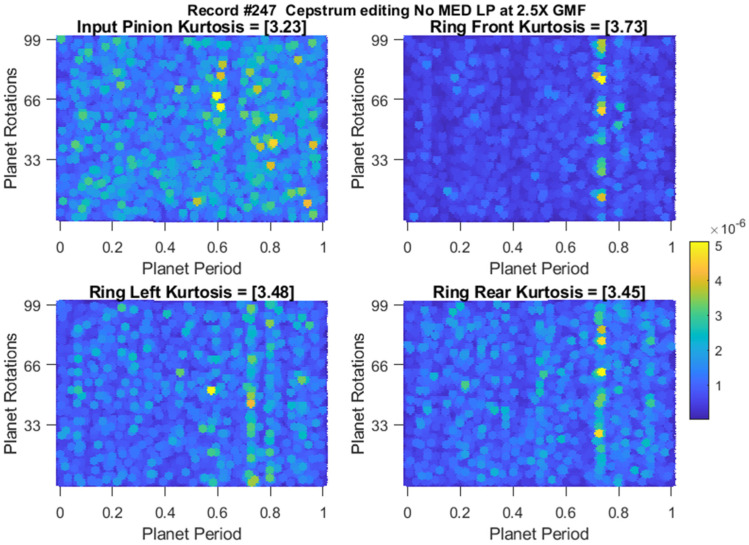
Squared envelope residuals obtained using cepstrum editing (no MED) at the earliest crack detection at record #247.

**Figure 22 sensors-24-02593-f022:**
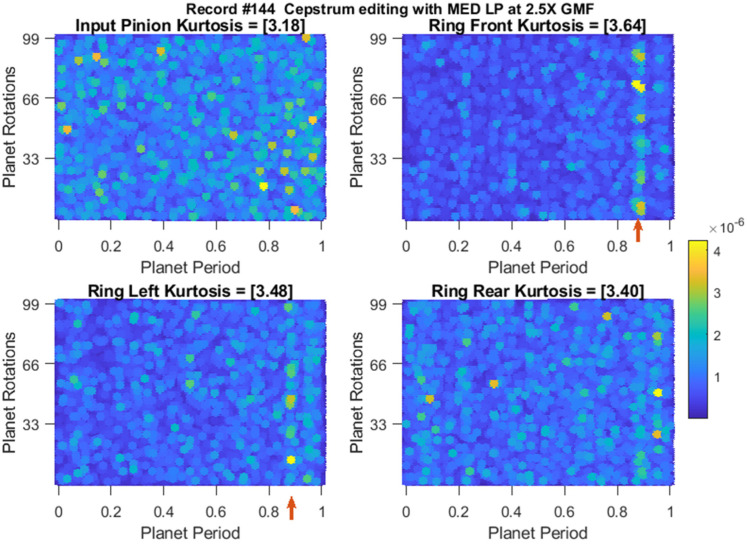
Squared envelope residuals obtained using cepstrum editing with MED at the earliest crack detection at record #144 (the red arrows indicates the impacts likely induced by planet-ring gear mesh).

**Figure 23 sensors-24-02593-f023:**
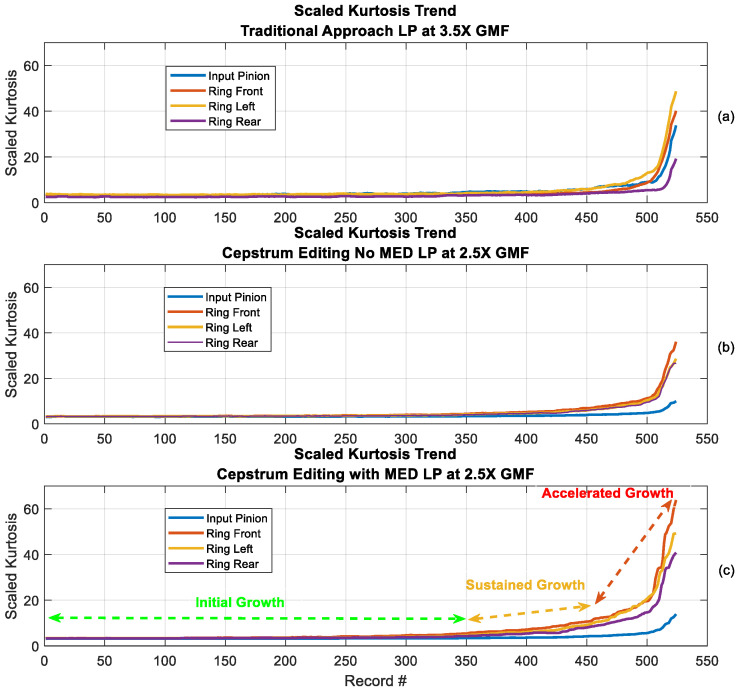
Scaled kurtosis trending: (**a**) Traditional approach with low-pass (LP) filter at 3.5× GMF (rolling median 9 samples), (**b**) Cepstrum editing with no MED (LP at 2.5 GMF: rolling median 12 samples), (**c**) Cepstrum editing with MED (LP at 2.5 GMF: rolling median 20 samples).

**Figure 24 sensors-24-02593-f024:**
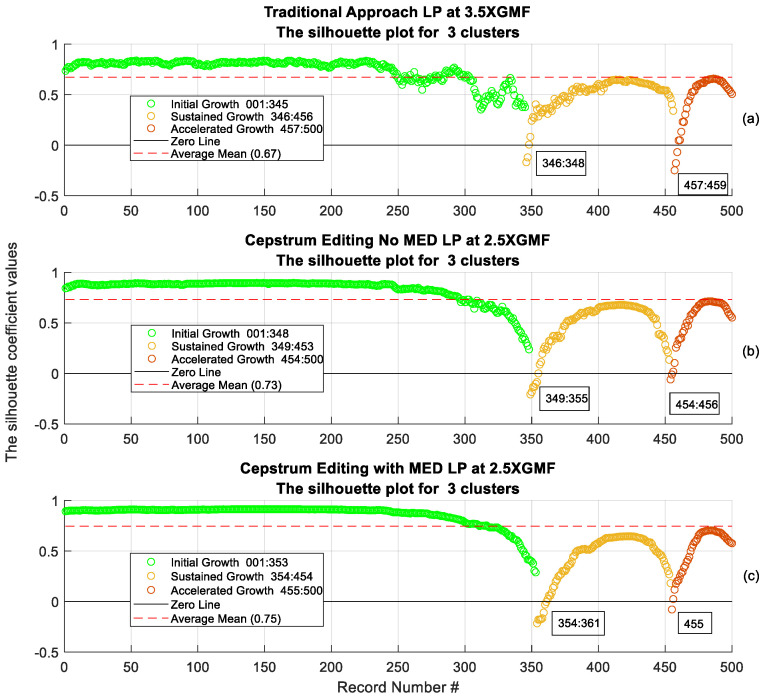
Assessment of k-means++ clustering using silhouette coefficient values: (**a**) Traditional approach, (**b**) Cepstrum editing, (**c**) Cepstrum editing with MED.

**Figure 25 sensors-24-02593-f025:**
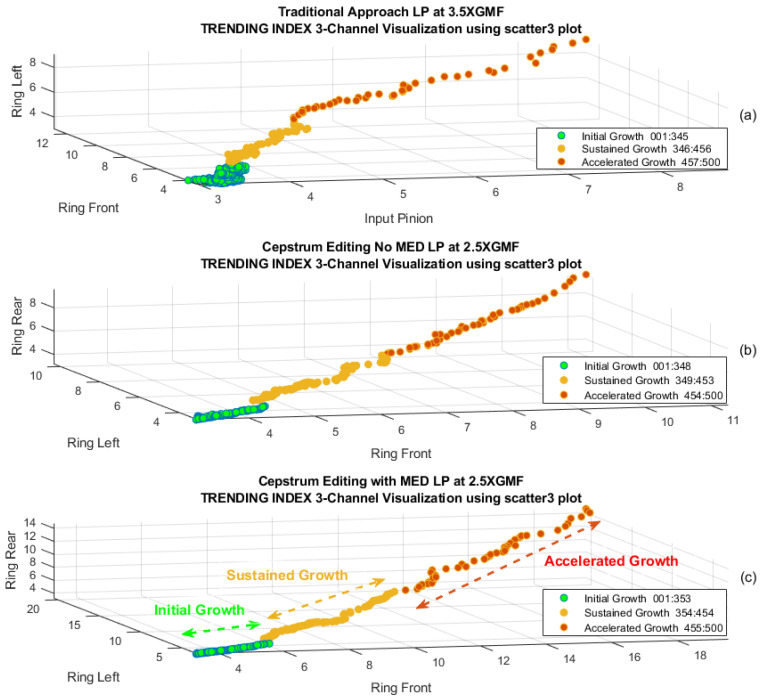
Three-dimensional visualization to identify accelerated crack growth: (**a**) Traditional approach, (**b**) Cepstrum editing, (**c**) Cepstrum editing with MED.

**Figure 26 sensors-24-02593-f026:**
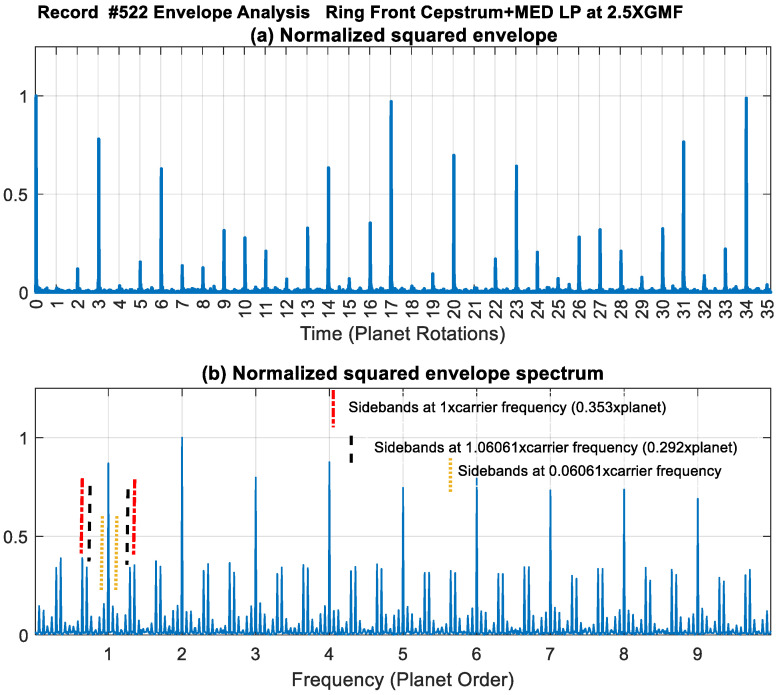
Envelope analysis of Ring Front sensor of signal obtained in Figure 17d (cepstrum-edited low-passed and Med filtered) (**a**) Normalized squared envelope (**b**) Normalized squared envelope spectrum.

**Table 1 sensors-24-02593-t001:** Key frequencies of interest (in Hz and planet orders).

	Frequency of Interest	Frequency (Hz)	Frequency (×Planet Orders)
Stage 1	Input Shaft	100	6.17
	Pinion/bevel gear mesh	1900	117.14
	Bevel gear shaft (sun gear)	26.76	1.65
Stage 2	Planet gear (relative to carrier)	16.22	1
	Sun gear (relative to carrier)	21.03	35/27 or 1.296
	Carrier (output shaft)	5.73	35/99 or 0.3535
	Planetary gear mesh frequency (GMF)	567.71	35 (or 99× carrier)
	Planet pass frequency (Number of planets times the carrier frequency)	3-planet 17.20 4-planet 22.94	3-planet 1.06 4-planet 1.41

**Table 2 sensors-24-02593-t002:** One load cycle (approximate durations).

Loading (%)	Load Input Torque (Nm)	Input Speed (RPM)	Duration * (Min)
50	152	6000	2
75	227	6000	2
100	303	6000	2
125	379	6000	24

* The first 9 cycles for the first notch had the actually durations of [3, 3, 3, 21] min at each load.

**Table 3 sensors-24-02593-t003:** Planet-ring impacts location (phase difference) with respect to a stationary sensor located at 0°.

*Planet-Ring* *Impacts*	*Spatial Location of the Planet Gear with Respect to the Fixed Sensor (0 to 360°)*	*Carrier Rotations*
** *1* **	*127.3*	*0.35354*
** *2* **	*254.5*	*0.70707*
** *3* **	* 21.8 *	* 1.06061 *
** *4* **	*149.1*	*1.41414*
** *5* **	*276.4*	*1.76768*
** *6* **	* 43.6 *	* 2.12121 *
** *7* **	*170.9*	*2.47475*
** *8* **	*298.2*	*2.82828*
** *9* **	* 65.5 *	* 3.18182 *
** *10* **	*192.7*	*3.53535*
** *11* **	*320.0*	*3.88889*
** *12* **	* 87.3 *	* 4.24242 *
** *13* **	*214.5*	*4.59596*
** *14* **	*341.8*	*4.94949*
** *15* **	*109.1*	*5.30303*
** *16* **	*236.4*	*5.65657*
** *17* **	* 3.6 *	* 6.01010 *

## Data Availability

The data used in this work were available to the public through a data challenge organized in the DSTG International HUMS conference. The data can be accessed using the following link: https://www.dst.defence.gov.au/our-technologies/helicopter-main-rotor-gearbox-planet-gear-fatigue-crack-propagation-test (accessed on 1 February 2024).

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
