# Peer review of "Helicopter Planet Gear Rim Crack Diagnosis and Trending Using Cepstrum Editing Enhanced with Deconvolution"

_sensors, 2024, doi:10.3390/s24082593_

Round 1
Reviewer 1 Report
Comments and Suggestions for Authors
1. Kindly avoid the term 'we' in the manuscript.
2. Enhance the Fig 1 quality.
3. How effective is the cepstrum editing (liftering) technique in isolating harmonics and sidebands of planet gears in the angular domain?
4. What advantages do low-pass filtering and minimum entropy deconvolution (MED) offer in enhancing fault features, specifically in the context of helicopter planetary gearboxes?
5. Elaborate on the real-world applicability of the proposed algorithm in detecting and trending fatigue cracks in helicopter planetary gear systems.
6. How well does the algorithm perform in scenarios with complex interactions between different gear sets and varying sources of vibration?
7. What specific aspects of the proposed algorithm contribute to its outperformance compared to traditional signal processing approaches?
8. Apply this algorithm to any benchmarking dataset and verify the results.
9. Are there limitations or challenges encountered during the comparative analysis that should be considered?
10. How accurately and reliably does the algorithm trend crack growth in planet gears over multiple load cycles?
11. Are there insights into the algorithm's ability to detect incipient fault features and potential areas for improvement?
12. What characteristics of the vibration dataset from the Helicopter Transmission Test Facility (HTTF) influenced the algorithm's performance?
13. Are there generalizability concerns when applying this algorithm to different datasets or gearbox configurations?
14. How can the insights provided by the proposed algorithm be integrated into maintenance strategies for helicopter planetary gear systems?
15. Are there considerations for implementing this algorithm in predictive maintenance frameworks?
16. How adaptable is the algorithm to different sources of vibration commonly encountered in helicopter operations?
17. How might advancements in signal processing and machine learning further enhance the capabilities of fatigue crack detection in gear systems?
18. Refer to this paper for gear condition monitoring
https://doi.org/10.3390/app122110917
19. Describe the hyperparameters tuned and performance metrics.
20. Revise the conclusion.
Comments on the Quality of English LanguageModerate editing of the English language is required
Reviewer 2 Report
Comments and Suggestions for Authors
- Figure 9 - Some of the labels are obscured by the plot grid.
- Overall, nice analysis and documentation. My only question/suggestion is additional discussion on the two choices related to implementation of the algorithm (tine width and inverse filter width). Specifically, if the intent of this research is toward real-time detection of faults (and not post-failure diagnostics) what information is needed a priori and what is needed from empirical tuning. This would be related to knowing the low pass filter length and calculating minimum error.
This article is well written and contains detailed and relevant data visualizations. The authors have a strong literature review and use a novel digital signal processing (DSP) technique on acceleration data taken from a helicopter gearbox containing a crack in the gearing components to show how the new analysis technique can be used to detect the fault from the acceleration data earlier than detectable by more traditional DSP methods. The information given on the analysis method is detailed and the methodology is scientifically sound. The method may prove to be useful in an expanded manor for different types of gearboxes and drive components for critical power train components in propulsion and safety.Author Response
See the attached file.

Round 2
Reviewer 1 Report
Comments and Suggestions for Authors
Congrats to the authors.
Comments on the Quality of English LanguageMinor editing of the English language is required